# Inverse modeling of turbidity currents using artificial neural network: verification for field application

Hajime Naruse[1] and Kento Nakao[2]

[1]Graduate School of Science, Kyoto University. Kitashirakawa Oiwakecho, Sakyo-ku, Kyoto, 606-8502 Japan
[2]Baseload Power Japan. SHINTORA-DORI CORE 3F, 4-1-1, Shimbashi, Minato-ku, Tokyo, 105-0004 Japan

**Correspondence:** Hajime Naruse (naruse@kueps.kyoto-u.ac.jp)

**Abstract.**

Although in situ measurements observed on modern frequently occurring turbidity currents have been performed, the flow characteristics of turbidity currents that occur only once every hundreds of years and deposit turbidites over a large area have not yet been elucidated. In this study, we propose a method for estimating the paleo-hydraulic conditions of turbidity currents from ancient turbidites by using machine learning. In this method, we hypothesize that turbidity currents result from suspended sediment clouds that flow down a steep slope in a submarine canyon and into a gently sloping basin plain. Using inverse modeling, we reconstruct seven model input parameters including the initial flow depth, the sediment concentration and the basin slope. A reasonable number (3,500) of repetition of numerical simulation using one-dimensional layer-averaged model under various input parameters generates a dataset of the characteristic features of turbidites. This artificial dataset is then used for supervised training of a deep learning neural network (NN) to produce an inverse model capable of estimating paleo-hydraulic conditions from data of the ancient turbidites. The performance of the inverse model is tested using independently generated datasets. Consequently, the NN successfully reconstructs the flow conditions of the test datasets. In addition, the proposed inverse model is quite robust to random errors in the input data. Judging from the results of subsampling tests, inversion of turbidity currents can be conducted if an individual turbidite can be correlated over 10 km at approximately 1 km intervals. These results suggest that the proposed method can sufficiently analyze field-scale turbidity currents.

## 1 Introduction

Turbidity currents are sediment-laden density flows that occur intermittently in deep sea environments (Talling, 2014). Turbidity currents are the main drivers of mass circulation processes in deep sea environments. In fact, estimating the flux of organic carbon transported and buried by turbidity currents is particularly necessary to understand the carbon cycle processes (Buscail and Germain, 1997; Heussner et al., 1999). In addition, the deposits of turbidity currents, i.e., turbidites, form a submarine fans

on the sea floor, which may function as large-scale hydrocarbon reservoirs (Kendrick, 1998; Yoneda et al., 2015), and are thus economically essential.

Through recent development of observational instruments, the velocity and flow depth of deep-sea currents can be measured directly (Hughes Clarke, 2016). Consequently, numerous records of turbidity currents have been reported at locations such as Squamish Bay, Canada (Hughes Clarke, 2016), Monterey canyon offshore California (Xu et al., 2004; Xu, 2010; Paull et al., 2018), and the Congo Submarine Channel (Vangriesheim et al., 2009; Azpiroz-Zabala et al., 2017). Surprisingly, these records revealed that turbidity currents occurs almost monthly in the modern submarine environments (Paull et al., 2018). On the contrary, the record in Squamish indicated seven events even in one day. However, the turbidites observed in outcrops and cores are deposited at intervals of 500-1000 years or longer. For example, Ishihara et al. (1997) investigated the deposits of the fore-arc basin (the Pliocene Awa Group) and reported that turbidite beds were deposited approximately once every 1200-1300 years. Clare et al. (2014) analyzed turbidites in the western Mediterranean Sea, off the northwest coast of Africa and the Apennines, and found that they were deposited with a frequency of about 1,400-36,000 years in all regions. In contrast to these geologic records, the recent field observations show that turbidity currents are not a rare event.

What caused the difference in observed frequency of modern turbidity currents compared with the records of ancient turbidites? One of the possibilities is that most of the turbidity currents observed in the present day may be very small in magnitude or are diluted, and leave little or no deposits in large areas. If this is the case, turbidites of several tens of centimeters thick observed in geologic records can be interpreted as deposits of extraordinarily large-scale events that occurred once every several hundred years. This hypothesis implies that turbidites in the strata resulted from low-frequency but very high-risk events such as large tsunamis and earthquakes (Goldfinger et al., 2003). In this case, we would have to assume that the velocity and concentration of turbidity currents obtained from in-situ observations are quite different from those of turbidite-forming currents in strata. Another possibility is that the areas where turbidity currents have been measured experienced very special conditions and that the frequency of turbidity currents will be significantly reduced even in those areas over long time scales. It is very difficult to determine which of these hypotheses is correct at this time because typical hydraulic conditions under which ancient turbidites were deposited have not been well understood. Although the characteristics of turbidity currents in natural environments have been elucidated rapidly through recent *in situ* observations of flow properties (Paull et al., 2018), the flow characteristics of turbidity currents that form actual submarine fans remain unknown.

The inverse analysis of turbidites in strata may fill the gap between the observations of turbidity currents and the geologic field observations of ancient turbidites. The reconstruction of past conditions by inverse analysis has been a major tool in several research fields including sedimentology and geomorphology. For example, several studies have reconstructed the magnitudes of past tsunamis from tsunami deposits (Jaffe and Gelfenbaum, 2007; Naruse and Abe, 2017; Mitra et al., 2020), and Rossano et al. (1996) estimated the behavior of pyroclastic flows using inverse analysis. If the hydraulic conditions of turbidity currents, such as velocity and concentration from turbidites can be reconstructed, it should be possible to verify whether turbidite beds in geologic records were deposited from flows of different scales or not by comparing the reconstructed values with the *in situ* observations.

However, no practical methodology for the inverse analysis of turbidity currents applicable on a field scale has yet been established. Early attempts to obtain hydraulic parameters of turbidity currents were based on the grain-size distribution of turbidites (Scheidegger and Potter, 1965; van Tassell, 1981; Bowen et al., 1984; Komar, 1985; Kubo, 1995) or on sedimentary structures (Harms and Fahnestock, 1960; Walker, 1965; Allen, 1982; Komar, 1985; Allen, 1991; Baas et al., 2000). The estimation of hydraulic conditions for turbidity currents based on grain size assumed that the flow is close to the criteria of suspension or the auto-suspension (Komar, 1985), but it has been emphasized that this assumption is highly problematic and leads to significantly different results compared with the actual hydraulic conditions for turbidity currents (Hiscott, 1994). Although the methods based on sedimentary structures can provide rough estimates of the conditions of a turbidity current, assumptions regarding the thickness of the flow are required (Ohata et al., 2017).

To obtain reasonable flow characteristics from turbidites, inverse analysis using a numerical model should be performed. Falcini et al. (2009) proposed a method for predicting the hydraulic conditions of turbidity currents from ancient turbidites and applied it to the Laga Formation in the Central Apennines, Italy. Their steady-state model was largely simplified to obtain an analytical solution of the model. However, most of the ancient turbidites are characterized by graded bedding (Bouma, 1962), which suggests non-steady waning nature of currents. Therefore, the applicability of this method should be quite limited to non-graded turbidites deposited from long-maintained flows. Conversely, Lesshafft et al. (2011) applied a direct numerical simulation model for the inversion of turbidite, whose application however to field-scale data is difficult because of the high calculation cost. Parkinson et al. (2017) proposed a method applicable to non-steady field scale flows by using a layer-average model as the forward model, which is potentially applicable to turbidites in outcrops. However, the flow conditions predicted from ancient turbidites were quite unrealistic in their study. They analyzed a turbidite in the Marnoso Arenacea Formation in the Appennine, and gave flow depth of 3950 m or 1.92 mm; both reconstructions are not acceptable as realistic conditions. These extremely large or small estimates may be due to oversimplification in their forward model or failure in the optimization of the input parameters. Nakao and Naruse (2017) were the first to successfully perform an inverse analysis of turbidites using a general non-steady layer-averaged model. Although their reconstruction of the hydraulic conditions of the turbidity current was reasonable, the computational load of the inverse analysis was high because they used a genetic algorithm for optimization. Thus, they were unable to repeatedly analyze various artificial or field data to test the validity and robustness of their inverse model. In addition, because of the high computational load, modifying their forward model to a more complex one in future would be difficult. These previous attempts suggest that a robust inverse model that can accept a more complex forward model is required to conduct inversion of turbidity currents from turbidites under realistic conditions.

Here, we propose a new methodology using an artificial neural network (NN) for obtaining flow characteristics of turbidity currents from their deposits (Fig. 1). NNs are machine learning systems that can be trained to perform very complex functions (Hecht-Nielsen, 1987). NNs have been used in a wide range of applications such as classification (Krizhevsky et al., 2012) or generative modeling (Sun, 2018). In recent years, this method has been widely applied also in the field of earth and planetary sciences (Laloy et al., 2018). Particularly, NNs are a powerful tool for high-dimensional regression of multiple variables with complex distributions (LeCun et al., 2015). In this study, we generate a non-linear regression model to estimate the hydraulic conditions of turbidity currents from the spatial distribution of bed thickness and grain size of turbidites using NN. If the

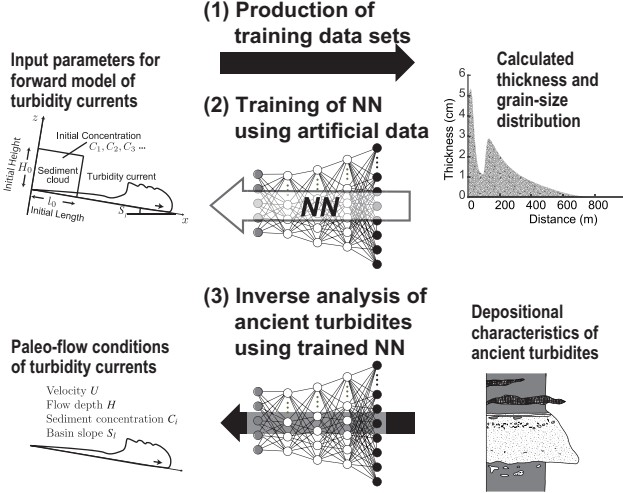

**Figure 1.** Schematic diagram of the inversion process of turbidity currents from deposits. The method is composed of three steps: (1) generation of training data sets by the forward model using random values for model input parameters, (2) trainng of NN based on the artificial data sets, and (3) application of the trained inverse model to unknown field data sets

regression is adequate, the NN can be used as an inverse model of turbidity currents. However, there are too few *in situ* measurements of hydraulic conditions of turbidity currents available. Although it is predicted that at least several hundred datasets of hydrological conditions and depositional characteristics are required to train NN, such frequent observation of turbidity currents that occur intermittently on the deep seafloor cannot be expected. Therefore, the method proposed in this study is designed to generate data of deposits from known conditions by numerical calculations of the forward model. In this case, the generation of training data can be completely parallelized, and therefore, any model that incur a high computational load can be implemented as a forward model.

In this study, we implement a NN-based inverse analysis and examine its effectiveness for turbidites at the field scale. The focus of this study is on rapidly decelerating sedimentary turbidity currents, and normally graded turbidites are considered to be deposited from such decaying flows. This approach was already proved to be effective for the inverse analysis of tsunami deposits (Mitra et al., 2020). However, the forward model used in their study was based on the assumption of quasi-steady flow, and thus our work is the first time to perform the inverse analysis using a neural network with completely unsteady flow. The success of the inverse analysis for turbidity currents, which exhibit quite different properties from those of tsunamis, would indicate the wide applicability of our inversion framework for event deposits.

## 2   Forward model description

Here we describe the formulation of the forward model used for producing training datasets for the inverse model (Fig. 2). This model is based on the model developed by Kostic and Parker (2006), which predicts the behavior of surge-type turbidity

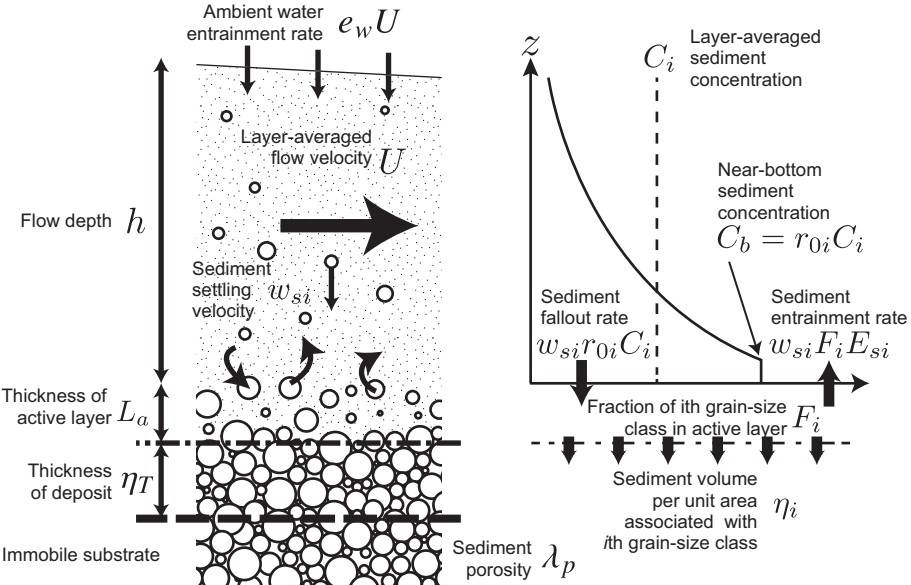

**Figure 2.** Explanation of model parameters. The turbidity current exchanges suspended sediment with the active layer ($L_a$ in thickness) on the top of the deposit ($\eta_T$ in thickness) by settling and entrainment. The volumetric rate of settling of the $i$th grain-size class of sediment is calculated from the basal sediment concentration $r_0 C_i$ multiplied by the sediment settling velocity $w_s i$. The sediment entrainment rate from the active layer is $w_{si} F_i e_{si}$, where $F_i$ is the volumetric fraction of the $i$th grain-size class in the active layer and $E_{si}$ is the unit dimensionless rate of sediment entrainment. The time variation of grain-size distribution in the active layer is computed in this model.

currents, but we modified it to consider sediment transport and deposition of multiple grain-size classes. The initial setting of the flows was set to be the lock-exchange condition, which assumes that the collapse of a rectangular-shaped cloud of sediment
suspension produces a turbidity current.

In a turbidity current flowing over hundreds of kilometers, (Luchi et al., 2018) suggested that the upper layer of the current is predicted to be continuously diluted while the lower layer remains highly concentrated, thus maintaining the current over long distances. Existing one-layer shallow water equation models are insufficient to reproduce such phenomena. The forward model of this study is not an exception.

However, the focus of this study is on rapidly decelerating sedimentary turbidity currents. Normally graded turbidites are considered to be deposited from such decaying flows. In this study, the distribution of turbidites is assumed to be limited to about several tens km at most, and the separation of the lower and upper layers that occurs in sustained turbidity currents after flowing tens of kilometers does not need to be considered when calculating such relatively small-scale turbidity currents. In fact, Kostic and Parker (2006), on which the forward model of this study is based, has been verified to reproduce turbidity
currents at experimental and small natural scales (e.g., Fildani et al., 2006). This suggests that the inverse model in this study is well suited to analyze a single bed of turbidites that generally exhibit normal grading in strata.

## 2.1 Layer-averaged equations

Let $t$ and $x$ be the time and bed-attached streamwise coordinates, respectively. Parameters $U$ and $h$ denote the layer-averaged flow velocity and the depth, respectively. The total sediment concentration is $C_T$. Here, we apply the following layer averaged conservation equations of fluid mass, momentum and suspended sediment mass of a turbidity current (Parker et al., 1986; Kostic and Parker, 2006):

$$\frac{\partial h}{\partial t} + \frac{\partial U h}{\partial x} = e_w U \tag{1}$$

$$\frac{\partial U h}{\partial t} + \frac{\partial U^2 h}{\partial x} = R g C_T h S - \frac{R g}{2}\frac{\partial C_T h^2}{\partial x} - C_f U^2 \tag{2}$$

$$\frac{\partial C_i h}{\partial t} + \frac{\partial U C_i h}{\partial x} = w_{si}(F_i e_{si} - r_0 C_i) \tag{3}$$

where $R(= \rho_s/\rho_f - 1)$ is the submerged specific density of the sediment ($\rho_s$ and $\rho_f$ are the densities of the sediment and the fluid), and $g$ is the gravity acceleration. $S$ is the slope, and $C_f$ denotes the friction coefficient. The right-hand side of the fluid mass conservation (Equation 1) considers the entrainment of ambient fluid to the flow, in which the empirical entrainment coefficient $e_w$ is applied. Equation 3 describes the mass conservation of the suspended sediment in the flow, which varies depending on the balance between settling and entrainment of the sediment from and to the active layer. In this model, the grain-size distribution of sediment is discretized to $N$ classes. The parameter $C_i$ denotes the suspended sediment concentration of the $i$th class. The model applies the active layer assumption, in which the grain-size distribution is vertically uniform in the bed surface layer (active layer) that exchanges sediment with suspended load (Hirano, 1971). $F_i$ indicates the fraction of the $i$th grain-size class in the active layer. The parameter $w_{si}$ denotes the settling velocity of the sediment particles in the $i$th class, and $r_0$ denotes the ratio of near-bed concentration to the layer-averaged concentration of the suspended sediment.

The mass conservation of the sediment in the active layer and the deposit (historical layer), respectively, takes respectively the form

$$\frac{\partial \eta_i}{\partial t} = \frac{w_{si}}{1-\lambda_p}(r_0 C_i - e_{si} F_i) \tag{4}$$

$$\frac{\partial \eta_T}{\partial t} = \sum \frac{\partial \eta_i}{\partial t}. \tag{5}$$

$$\frac{\partial F_i}{\partial t} + \frac{F_i}{L_a}\frac{\partial \eta_T}{\partial t} = \frac{w_{si}}{L_a(1-\lambda_p)}(r_0 C_i - F_i e_{si}). \tag{6}$$

where $\eta_i$ denotes the volume per unit area of the $i$th grain size class, and $\eta_T$ is the total thickness of the deposit. $L_a$ denotes the thickness of the active layer, which is assumed to be constant, for simplicity. The parameter $\lambda_p$ denotes porosity of the active layer and the deposit (0.4 in this study), and $e_{si}$ is an empirical coefficient for sediment entrainment of the $i$th class from the active layer. Equation 4 describes the mass conservation of the $i$th class sediment in the bed, and rate of the bed aggradation is obtained by summation of accumulation rates of all grain-size classes (Equation 5). Equation 6 considers the

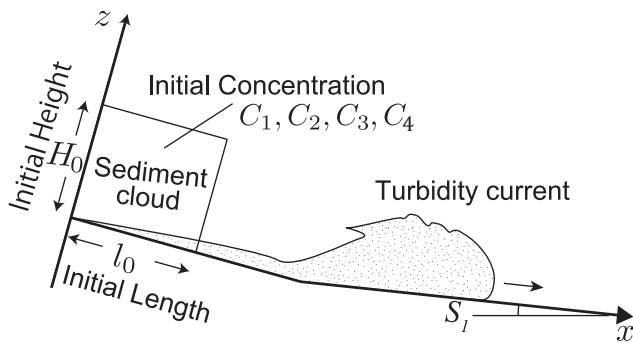

**Figure 3.** Model input parameters. The initial conditions of turbidity current is assumed to be the suspended sediment cloud that is $H_0$ and $l_0$ in height and length, respectively. The initial sediment concentrations $C_1$ to $C_4$ and the basin slope $S_l$ are to be specified for calculation. These seven input parameters are subject to be reconstructed by inverse analysis.

temporal development of the grain size distribution in the active layer, where the time development of the total bed thickness $\eta_T$ is obtained by summation of the right-hand side of Equation 4 for all grain-size classes.

For computation, to solve the Equations 1–6, empirical relations are required for the parameters: $w_{si}$, $r_0$, $C_f$, $L_a$, $e_w$, and $e_{si}$. Here we applied the formulation of Dietrich (1982) for obtaining the settling velocity $w_{si}$. The ratio of near-bed to layer-averaged concentrations $r_0$ and the bed friction coefficient $C_f$ are fixed to be 2.0 and 0.004, for simplicity (Garcia, 1990). The active layer thickness $L_a$
is assumed to be constant (0.003 m). Regarding the entrainment coefficients of ambient water and basal sediment $e_w$ and $e_{si}$, we applied formulations proposed by Parker et al. (1987) and Garcia and Parker (1991) respectively.

For computational efficiency and numerical stability, a deformed grid approach was adopted to solve Equations 1–3. In this transformed coordinate, the propagating flow head was fixed at the downstream boundary using a Landau transformation (Crank, 1984). The tail of the flow was also fixed at the upstream end of the calculation domain, and thus the grid spacing in
the dimensional coordinate space was continuously stretched during calculation, whereas that in dimensionless space remained constant. This scheme was based on Kostic and Parker (2006), and more details regarding the numerical implementation were given by Nakao and Naruse (2017).

### 2.2 Model input parameters and topographic settings

In this study, a turbidity current was assumed to occur from a cloud of suspended sediment (height: $H_0$, length $l_0$). The initial
flow velocity was set to 0, and the sediment of the $i$th grain-size class was considered to be initially homogeneously distributed in the suspension cloud at the concentration $C_i$ (Fig. 3). The suspended sediment cloud was located at the upstream end of the calculation domain, where the slope gradient was 0.1. This steep slope extended for 5.0 km and transited to a gently sloping basin plain (gradient is $S_l$) in the downstream region. Total length of calculation domain was 100 km. In summary, the number of initial conditions required for the forward model calculation was three ($H_0$, $l_0$ and $S_l$) plus number of grain-size classes
($C_i$).

## 3  Inverse modeling by deep learning NN

In this study, numerical simulation of a turbidity current is repeated under various random initial conditions to produce a data set of the characteristic features of turbidites. Then, this artificial dataset of turbidites is used for supervised training of a deep learning NN. The values of the turbidites characteristics, i.e., distribution of volume-per-unit-area of all grain size classes, in the training data set are input to the NN, and the estimated initial conditions (e.g., initial flow height and concentration) of the turbidity current is obtained from the output nodes of NN. The output values of the NN are compared with the true conditions. The optimization of weight coefficients of NN is then conducted to reduce the mean square of the difference between the true conditions and the output values of the NN. If the number of training datasets is sufficiently large, the trained NN should be able to estimate the paleo-hydraulic conditions from the data of the ancient turbidites (Fig. 1). In other words, an empirical relationship with numerical results and the model input parameters are explored in this method, and the discovered relationship is used for inverse modeling of turbidity currents.

The local conditions of a turbidity current (velocity, concentration, etc.) at any locations and time can be estimated from the reconstructed initial conditions. The flow parameters are obtained by calculating the time evolution of the forward model from the initial conditions. In this way, we can obtain the behavior of the flow with a relatively small number of parameters. This approach has already been tried successfully by Lesshafft et al. (2011), and Falcini et al. (2009) also reconstructed flow conditions of turbidity currents by obtaining boundary conditions of the model.

The details of these procedures are described below.

### 3.1  Production and preprocessing of training and test data sets for supervised machine learning

We conducted iterative calculations using the forward model and accumulated data to train and validate the inverse model. To investigate the appropriate amounts of data for training the inverse model, we conducted 500–3500 iteration of the forward model calculations. To verify the performance of the trained model, 300 test data sets were also generated numerically, independent of the training data.

Model input parameters that are subject to inversion are required to produce the training and test data by the forward model calculation (Fig. 3). In this study, the model inputs are the initial flow height $H_0$, the initial flow length $l_0$, the initial sediment concentration for the $i$th grain size class $C_i$, and the basin slope $S$. These model parameters are generated as uniform random numbers within a certain range, and their range is changed according to the target of the inverse analysis. Since this study is aimed at field-scale analysis, the following ranges are chosen. Both initial depth and length of suspended cloud range from 50 to 600 m. The sediment concentration for each grain size class ranges from 0.01% to 1.0%. The number of grain size classes $N$ is four, and the representative grain diameters are 1.5, 2.5, 3.5 and 4.5 phi. The inclination of the basin plain where the turbidites are expected to form ranges from 0 to 1.0%.

Each run of the forward model calculation is initiated with the given model input parameters, and is terminated when the flow head reaches the downstream end or sufficiently long time period ($1.2 \times 10^5$ s.) has elapsed. As a result of the calculation, the forward model outputs the volume-per-unit-area of sediment for all grain size classes over the 100 km-long calculation

domain. The inverse model estimates the model input parameters from the resultant spatial distribution of the granulometric
characteristics of the deposits. However, in natural outcrops, it is unlikely that the entire distribution of the turbidite beds
would be exposed. Therefore, we limit the length of the sampling window in the calculation domain, and only the sediment
data contained in this window is extracted for both training and testing. The upstream end of the sampling window was set at
the transition point between the steep slope and the basin plain (5 km from the upstream end), and the length of the window
varies from 1 to 30 km to evaluate the data interval required for the inverse analysis.

Before the model input parameters are input to NN, all values are normalized between 0 and 1 using the following equation:

$$I_i^* = \frac{I_i - I_{min}}{I_{max} - I_{min}} \tag{7}$$

where $I_i^*$ and $I_i$ denote the $i$th normalized and original input parameters, respectively. $I_{\mathrm{max}i}$ and $I_{\mathrm{min}i}$ are the maximum and
minimum values used for generating the $i$th input parameter, respectively. This min-max normalization is applied to consider
all parameters at equal weights because the range of the initial flow conditions is significantly different between them.

## 3.2 Structure of NN

The artificial NN is used as the inverse model to reconstruct flow conditions from the depositional architecture. We input the
spatial distribution of volume-per-unit-area of multiple grain size classes of a turbidite in the NN, which outputs the values
of the flow initial conditions and the basin slope. In this study, we use a fully connected NN that has four hidden layers. The
volume-per-unit-area of $N$ grain-size classes of sediment deposited on $M$ spatial grids in the sampling window is given to the
input nodes of the NN. Thus, the total number of the NN input nodes is $N \times M$. The number of nodes in all hidden layers is
set to 2000 in this study.

The Rectified Linear Unit (ReLU) activation function is adopted for all NN layers (Nair and Hinton, 2010; Glorot et al.,
2011). The ReLU is the half-wave rectifier $f(z) = \max(z, 0)$. Compared with other smoother non-linearities, such as $\tanh(z)$
or $1/(1 + \exp(-z))$, the ReLU typically learns much faster in NN with multiple layers (Glorot et al., 2011), and thus it allows
to train a deep supervised network without unsupervised pre-training (LeCun et al., 2015).

The NN is expected to output the model input parameters (i.e., the initial flow conditions and the basin slope), and therefore,
the number of nodes in the output layer is equal to the number of input parameters for the forward model, which is seven here
(the initial flow length, depth, sediment concentrations and the basin slope).

## 3.3 Training the inverse model

To develop the inverse model, supervised training is conducted using the artificial dataset produced by the forward model
calculation. First, the artificial dataset is randomly split into training and validation datasets to detect overfitting during the
training process. The ratio of the validation dataset is set to 0.2 so that 80% of the artificial dataset is used for training. The
model input parameters used for producing training and validation sets were regarded as the teacher data to train and evaluate
the model.

Methodology applied for training the NN is as follows. The mean squared error (MSE) is adopted as the loss function because the supervised training of NN in this study is classified as a regression problem (Specht, 1991), and MSE is a common loss function for regression (Bishop, 2006; Hastie et al., 2009; Shalev-Shwartz and Ben-David, 2014). Before training, all weight coefficients of NN are randomly initialized using the Glorot uniform distribution (Glorot and Bengio, 2010). The backpropagation algorithm (Rumelhart et al., 1986) is used to calculate the derivative of this error metric for each connection

between the nodes, and the stochastic gradient descent method (SGD) with Nesterov momentum (Nesterov, 1983) is used for optimizing the weight coefficients of NN to minimize the difference between the model predictions and the teacher datasets. Other optimization methods, such as AdaGrad (Duchi et al., 2011), RMSprop (Tieleman and Hinton, 2012) and AdaDelta (Zeiler, 2012), have been tested, but SGD shows the best performance in this case. Dropout regularization (Srivastava et al., 2014) is applied for each epoch to reduce overfitting and to improve the generalization ability of the NN. One training epoch,

which refers to one cycle through the full training dataset, is repeated until the loss function of the validation dataset converges to a constant value. These methods are all implemented in Python with the library Tensorflow 2.1.0 (Raschka and Mirjalili, 2019), and the calculations are conducted using GPU NVIDIA GeForce GTX 2080 Super with libraries CUDA 11.0 and CuDNN 7.0.

Several hyperparameters should be specified for the training of NN. Specifically, the dropout rate, the learning rate, the batch

size, the number of epochs, and the momentum are adjusted manually after repeated trial and error. To perform an optimization calculation with SGD, the batch size and the learning rate were set to 32 and 0.02, and the value 0.9 was chosen for the momentum. Dropout rate for regularization was 0.5.

### 3.4 Testing the inverse model

The performance of the inverse model is tested using a set of 300 data that are produced independently of the training and

255 validation datasets. The inversion precision for each model input parameter is evaluated by the root mean square error (RMSE) and the mean absolute error (MAE) of the prediction. These error metrics are computed for both raw and normalized values with true values, and used to evaluate the model. Moreover, the bias of prediction (i.e., the mean deviation of the model predictions from the true input parameters) is used describe the accuracy of the inversion.

Three additional tests are conducted for verifying the robustness of the inverse model that is significant for the applicability

of the model to field datasets. The results of these tests are evaluated by the average of the normalized RMSE, which is defined as:

$$\text{RMSE} = \sqrt{\frac{1}{JK} \sum_J \sum_K \left(\frac{I_{\text{p}jk} - I_{jk}}{I_{jk}}\right)^2} \tag{8}$$

where $I_{\text{p}jk}$ and $I_{jk}$ denote the predicted and the original values of the $j$th model input parameter for the $k$th test dataset, respectively. $J$ and $K$ are the numbers of the model input parameters and the test data sets.

First, noise is artificially added to the test data to evaluate the robustness of the inversion results against the measurement error. Under natural conditions, measurement errors in the thickness and grain size analysis of turbidites as well as the local topography affect these results. If the results of the inverse analysis change significantly due to such errors, it means that our method is not suitable for application to field data. To investigate this, we apply normal random numbers to the volume per unit area at each grid point in the training data at various rates, and we observe how much influence the noise has on the inverse analysis results.

The second test on the inverse model is to perform a subsampling of the grid points in the training data. Outcrops are not continuous over tens of kilometers, so that the thickness and the grain size distribution of a turbidite in the interval between outcrops can only be obtained by interpolation. To simulate this situation, the grid points in test datasets are randomly removed in this test, and the volume-per-unit-area at the removed grid points is linearly interpolated. By varying the rate at which grid points are removed, this test also allows us to estimate the average interval of the outcrops that are necessary for conducting the inverse analysis. That is, if 90% of the grid points set at 5 m intervals are removed, and the inverse analysis is conducted on the remaining 10%, the average distance between the grid points is 50 m. Estimating the outcrop spacing requires obtaining reasonable results of inverse analysis before applying it to the actual field.

Finally, the influence of the length of the upstream slope was examined. In this study, it is assumed that a steep slope (10 %) of a submarine canyon with a length of 5 km exists upstream, and a basin plain with a gentle slope exists downstream of the steep slope. Although the topography and deposits of the upstream slope are not the subject of the inverse model analysis, the length of the slope potentially affect the results of the inverse analysis. As a test, we set a slope of 10 km length instead of 5 km upstream, and deposited a turbidite bed from the turbidity current flowing down from the uppermost part of the slope. The turbidite was then analyzed using a model trained on the assumption of 5 km slope to compare the reconstructed values with the original conditions.

## 4 Results

### 4.1 Properties of artificial data sets of turbidites

Here, we describe the properties of turbidite artificial data generated for training and testing the inverse model. Several artificial datasets of turbidites are produced using a 1-D shallow water equation model. Figure 4 exhibits examples of the calculated spatial distribution in bed thickness and grain size of turbidites deposited in the region of the basin plain. Most beds exhibit the typical "top-hat" or "core and drape" shape of turbidites Hirayama and Nakajima (1977); Talling et al. (2012); Pantopoulos et al. (2013), where turbidite beds become thicker in the upstream part of the basin and then thin rapidly from their peak of thickness. Thereafter, beds continue over a long distance, gradually decreasing in thickness (Fig. 4). At the same time, the grain size gradually becomes finer downstream. The maximum thickness of beds is 1.27 m on average (standard deviation $\sigma = 1.65$ m), and the mean value of the area where sediments with a thickness greater than 1 cm are distributed is 42.0 km ($\sigma = 15.7$ km). Each bed is composed of four grain size classes. All distributions of the volume-per-unit-area of the grain size classes

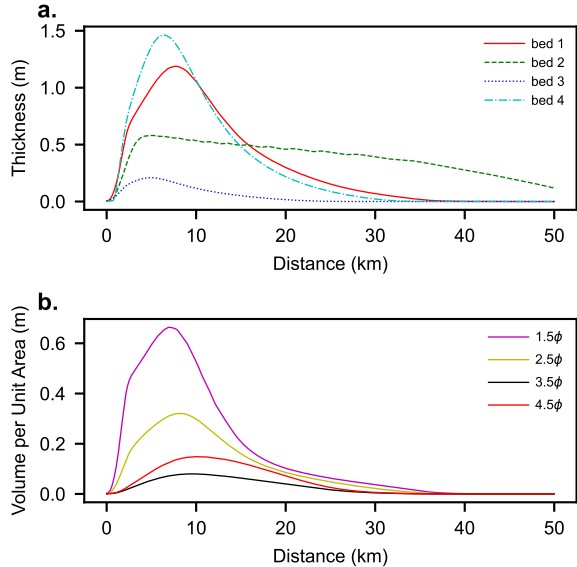

**Figure 4.** Examples of turbidites calculated by the forward model. **a**. Spatial distributions of bed thickness. Four beds (Bed 1–4) were plotted as examples. **b**. Spatial distribution of the volume-per-unit-area for each grain size class in Bed 1 (Fig. 4a).

are still "top hat" shaped (Fig. 4b), but the depositional center and the amounts of deposition are different for each classes depending on their size.

## 4.2  Results of training

We trained the NN inverse model with various numbers of artificial data and lengths of the sampling window, and the best result in terms of the value of the loss function for the validation sets and the practical usage of the model can be obtained with 3500 training data sets and 10 km-long sampling window (Fig. 5). Results with less than 2000 training data sets produce a discrepancy in the loss function between the training and the validation sets, indicating overlearning of the NN. Conversely, when the number of data sets exceeds 2000, the loss function of the validation set is slightly less than the value of the training

set. As the number of training data increases, the resultant values of the loss function improve. However, when the number of data exceeds 2500, the improvement of values of the loss function became not so rapid. Regarding the distance of the sampling window, the training results are not stable when the sampling window is shorter than 5 km (Fig. 5). On the other hand, the training results are stable when the window length is longer than 10 km, and the results gradually improve as the window length increases. However, extending the window length from 10 km to 30 km results in little improvement of the loss function. We

do not fully understand why the results are not stable for sampling windows shorter than 5 km, but it probably indicates that the training results fall into a local optimum solution depending on the initial values of the weight coefficients of the neural network (given by random numbers) due to incomplete information. In any case, the loss function is very good (less than 0.01),

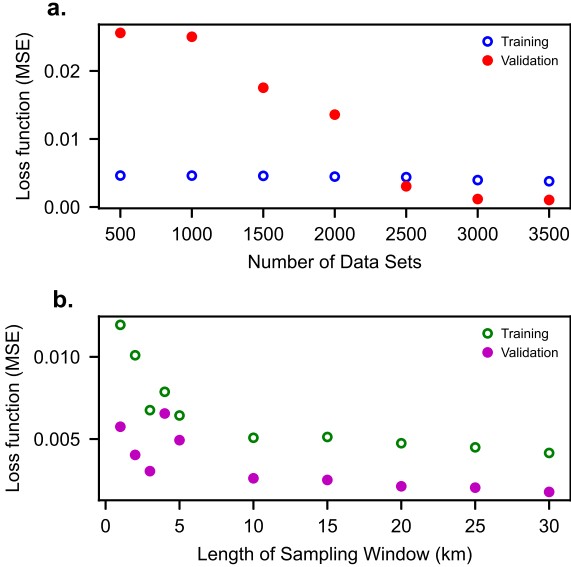

**Figure 5.** Results of training of the NN with different numbers of training data sets and lengths of the sampling window.

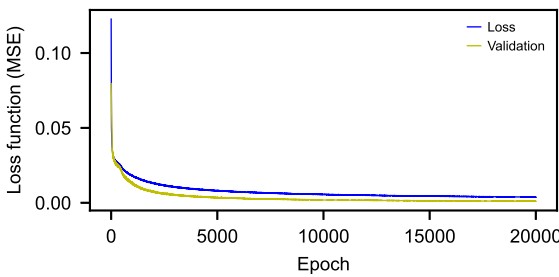

**Figure 6.** Training History of the NN. 3500 datasets and 10 km-long sampling window were used for this training.

so that even turbidites that can be tracked for less than 5 km are likely to give good results if the outcrop spacing is sufficiently narrow and detailed observation of beds is possible.

Hereafter, we further investigate the performance of the inverse model trained on a 3500 dataset with a 10 km-long sampling window. The history of training indicates that the values of the loss function improved significantly in the first 1000 epochs, and the results are improved up to 15,000 epochs (Fig. 6). Eventually, saturation is reached at approximately 20,000 epochs. The resultant loss function (i.e., the MSE of prediction) is $3.78 \times 10^{-3}$ for training sets and is $1.03 \times 10^{-3}$ for validation sets.

**Table 1.** Errors and bias of the predicted parameters. Prediction errors are exhibited by the root mean squared error (RMSE) and the mean absolute error (MAE), and the mean bias is also described. Normalized values of RMSE, MAE and mean bias by true values are also shown.

| | $R^2$ | RMSE | RMSE (normalized) | MAE | MAE (normalized) | Mean bias | Mean bias (normalized) |
|---|---|---|---|---|---|---|---|
| Initial height | 0.99 | 18.97 m | 8.55 % | 14.81 m | 5.96 % | -12.93 m | -5.18 % |
| Initial length | 0.99 | 15.82 m | 7.53 % | 12.09 m | 4.92 % | -2.33 m | -2.06 % |
| $C_1$ | 0.99 | 0.02 % | 12.91 % | 0.02 % | 6.00 % | -0.01 % | -4.44 % |
| $C_2$ | 0.99 | 0.02 % | 15.57 % | 0.02 % | 7.67 % | -0.01 % | -4.29 % |
| $C_3$ | 0.99 | 0.02 % | 13.03 % | 0.02 % | 6.39 % | -0.00 % | -2.49 % |
| $C_4$ | 0.99 | 0.03 % | 13.71 % | 0.02 % | 6.67 % | -0.01 % | -4.21 % |
| $S_l$ | 0.98 | 0.03 % | 19.56 % | 0.03 % | 11.67 % | 0.03 % | 11.45 % |

**Table 2.** The predicted and the true parameters used for an example of calculation for time evolution of the flow characteristics.

| | Initial height (m) | Initial length (m) | $C_1$ | $C_2$ | $C_3$ | $C_4$ | $S_l$ |
|---|---|---|---|---|---|---|---|
| True input parameters | 484.41 | 318.18 | 0.17 | 0.05 | 0.95 | 0.74 | 0.23 |
| Estimated parameters | 454.67 | 301.73 | 0.18 | 0.02 | 0.93 | 0.73 | 0.25 |

### 4.3 Precision and accuracy of inverse analysis

Using 300 test data sets, the performance of the inverse model trained with 3500 data sets and 10 km-long sampling window is evaluated. The estimated parameters are matched well with slight deviations (Figs. 7, 8; Table 1). $R^2$ values are beyond 0.98 for all parameters. Particularly good agreement is obtained for the estimates of the initial height and the length of the suspended sediment cloud. Values of the normalized RMSE and MAE for these parameters are less than 9 % and 6 %, respectively. The sediment concentration is also precisely estimated. the normalized RMSE for the sediment concentration ranges from 12 to 16

%, which corresponds to only 0.02–0.03 volumetric %. The prediction for the basin slope shows relatively large errors (RMSE is close to 20 % and MAE is 11.7 %), but these errors correspond to only 0.03 % of slope. Focusing on the bias of the estimates, all estimated values except for the basin slope tend to be slightly smaller, whereas the predicted values of the basin slope tend to be larger (Fig. 8). The values of the bias, however, range only from 2 to 12% of the original value.

The forward model is calculated again using the reconstructed values to examine the influence of the estimation error of the

model input parameters on the predicted flow behavior (Fig. 9). The chosen test values deviate from the true conditions as indicated by the RMSE value 0.27 (Table 2), but the time evolution of the flow characteristics agree very well with those calculated from the true values (Fig. 9). When comparing the velocity and concentration of the flow at 10 km from the upstream end, the discrepancy between calculation results using reconstructed and original parameters is less than 5% for both parameters.

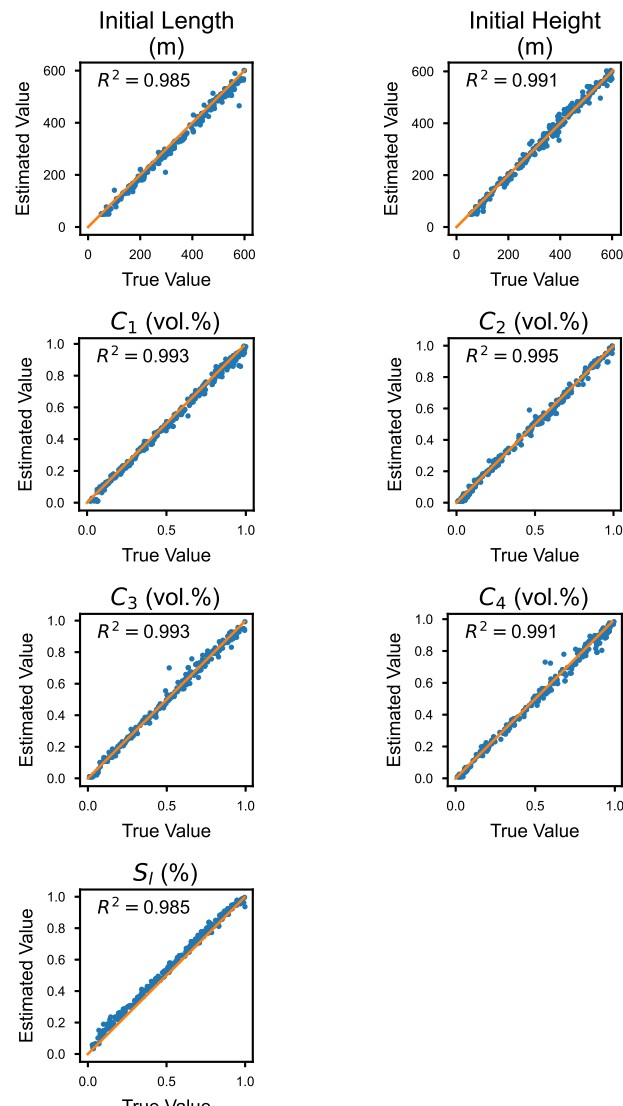

**Figure 7.** Result of the inverse analysis compared with the true parameters. The x-axis represents the value of the true parameter and the y-axis represents the value of the estimated parameter. The orange lines show that the two values are in a 1:1 relationship, and thus the plots on this line indicate that the prediction is perfectly consistent with the true value.

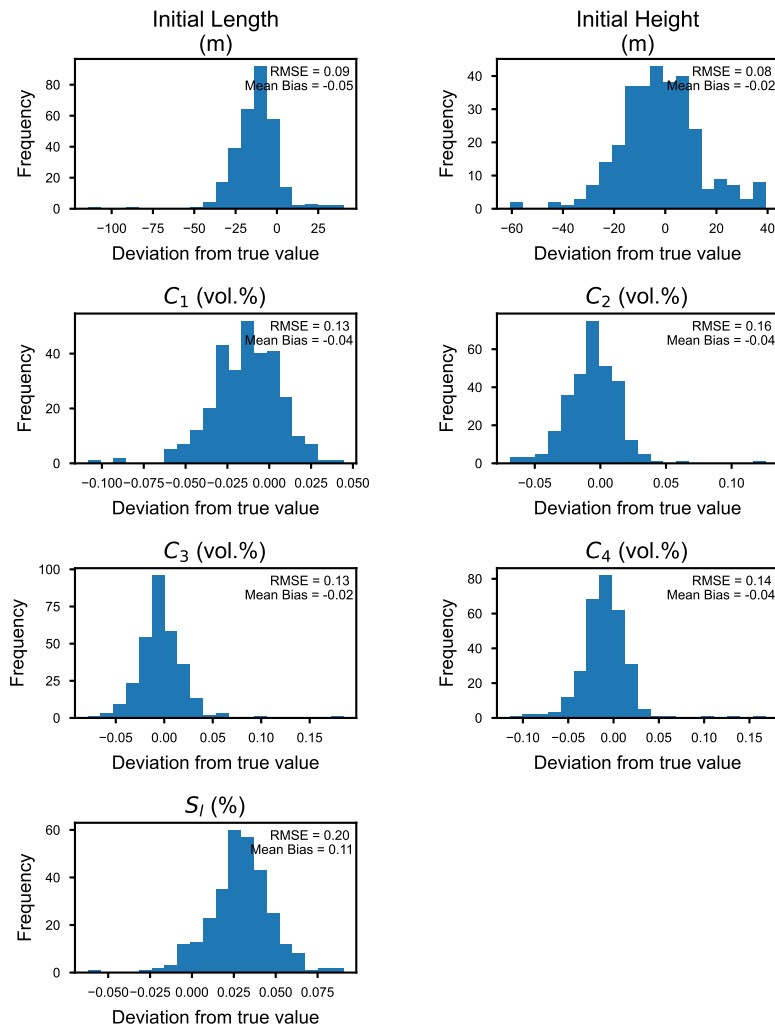

**Figure 8.** Histograms indicating the deviation of the predicted values from the true values.

## 4.4 Tests for robustness against noise and subsampling on input data

The test data with various amounts of normal random values are analyzed to verify the robustness of the inverse model. Consequently, even when the standard deviation of the normal random numbers given as measurement errors was set to approximately 200% of the value of the original data, only a small effect is observed in the normalized RMS of the results of the inverse anal-

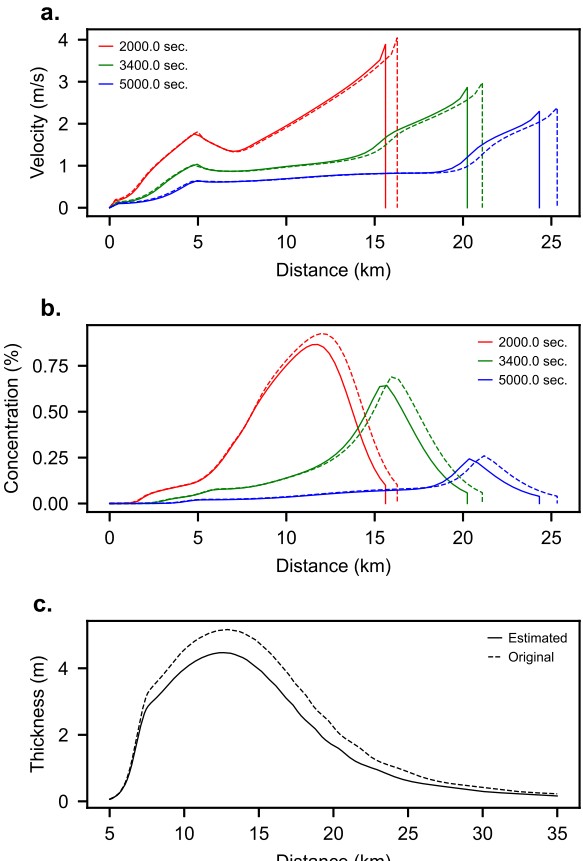

**Figure 9.** Example of forward model calculation with reconstructed and true parameters. The solid line indicates the calculation result using the predicted parameters, and the dashed line exhibits the results using the true parameters. a. Velocity distribution at 2000, 3500 and 5000 seconds after the flow initiation. b. Total sediment concentration at 2000, 3500 and 5000 seconds after the flow initiation. c. Spatial distribution of bed thickness at 25,000 seconds after the flow initiation.

ysis (Fig. 10). The RMS values gradually increase when the standard deviation of errors exceeds 50%, but there is no rapid increase in the RMSE of the results at any particular threshold.

Similarly, using subsampling data obtained by extracting some of the spatial grids from the original data, we conducted an inverse analysis of the test datasets. The results show that there is little influence on the RMSE values of the inverse analysis of the test datasets when the sampling rate of grids is greater than 1 % (Fig. 11). The RMSE values gradually increased when the sampling rate falls below 1 %, and RMSE becomes extremely high when the rate drops below 0.4%.

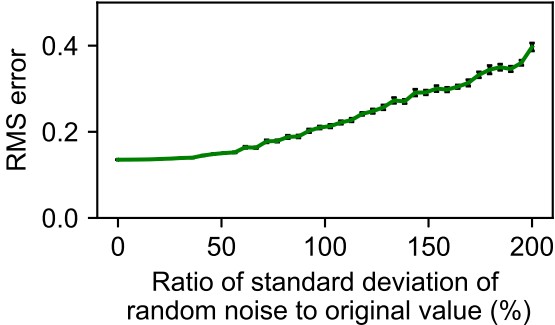

**Figure 10.** Result of inverse analysis of the test data sets with artificial noise. The values of RMSE are averaged over 20 times iterations. Error bars indicate standard errors of RMSE values.

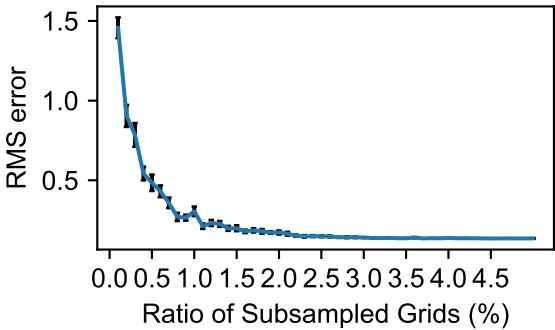

**Figure 11.** Results of the inverse analysis for the subsampled test data sets. The values of RMSE are averaged over 20 times iterations. Error bars indicate standard errors of RMSE values.

### 4.5 Tests for influence of length of the upstream slope

Here, a turbidite deposited in a different topographic setting was analyzed to determine the influence of the topographic assumptions to inversion results. The slope of 10 km instead of 5 km was set at the upstream end of the calculation domain. The initial conditions for this test assuming a 10 km slope were a suspended sediment cloud of 359 m high and 227 m long, with concentrations of 0.13 %, 0.15 %, 0.38 %, and 0.65 % for the four grain size classes, respectively. The gradient of the downstream slope was set to be 0.69 %.

As a result, the initial conditions estimated by the inverse model trained on the assumption of 5 km upstream slope were a suspended sediment cloud of 117 m high and 587 m long, with concentrations of 0.33 %, 0.38 %, 0.48 %, and 0.53 % for each grain size class, and a downstream slope was estimated to be 0.96 %. Then, these initial conditions were given to the forward

model to calculate their time development, and the obtained parameters were compared on a basin plain where the turbidite was deposited (Fig. 12).

The results exhibited that the model with a 5 km slope predicted relatively close values to the original results for the flow velocity (Fig. 12). Both the model with a 5 and 10 km slope calculated velocities that were approximately 3 m/s in maximum over the basin plain and gradually decelerated downstream. However, because the slope length is different, the time to reach each point on the basin plain differs greatly.

In contrast, the concentration of the turbidity current was significantly overestimated in the model reconstruction assuming

a 5 km slope (Fig. 12). When the flow reaches the downstream gentle slope at about 10 km, the original turbidity concentration was about 0.2 % at maximum, while the restored value was closer to 0.5 %. As a result, the thickness of turbidite estimated from the reconstructed initial values was also thicker than the original values.

## 5   Discussion

### 5.1   Performance of inverse model

The performance of the inverse model for turbidity currents is evaluated using the test data set, implying that this model can accurately reconstruct the flow characteristics of the turbidity currents from the spatial distribution of the thickness and grain size of turbidites (Figs. 7 and 8). The biases in the values reconstructed from the true input parameters are also very small and thus should not pose a serious issue when the method is applied to actual field data.

The inverse model not only reconstructed the initial conditions of turbidity currents accurately, but also the predicted time

evolution of the flow behavior was sufficiently accurately and precisely. In the results of the forward model calculations using the predicted model input parameters that are relatively deviate from the true values (Table 2), the time evolution of the velocity and the thickness of the flow does not deviate significantly from the results using the true values (Fig. 9).

Turbidity currents have a mechanism called the self-acceleration, which is caused by erosion and associated increase of the flow density (Parker et al., 1986; Naruse et al., 2007; Sequeiros et al., 2009). Therefore, even slight differences in the initial

conditions of the flow can lead to very different results of the time evolution of the flow parameters. However, the results of this test imply that the accuracy of the inverse analysis in this study is enough to prevent to cause such a drastic change in the flow behavior.

The relationship between turbidity currents and characteristics of turbidites is nonlinear. Especially when the flow is self-accelerating, a small difference in the initial conditions can result in very different sedimentary characteristics. This means that

it is easy to find the initial conditions of the flow by inverse analysis, because even if the characteristics of the deposits are very different, the initial conditions of the flow should not be so different. Thus, the inverse results in this case are expected to be robust even if there are some measurement errors in characteristics of deposits. In other words, there is a tradeoff between the robustness of the forward and inverse modeling.

This property of the inversion can be understood when we consider the opposite case. If the initial conditions of the flow

are different but the characteristics of the turbidites are exactly the same, it is impossible to estimate the flow conditions from

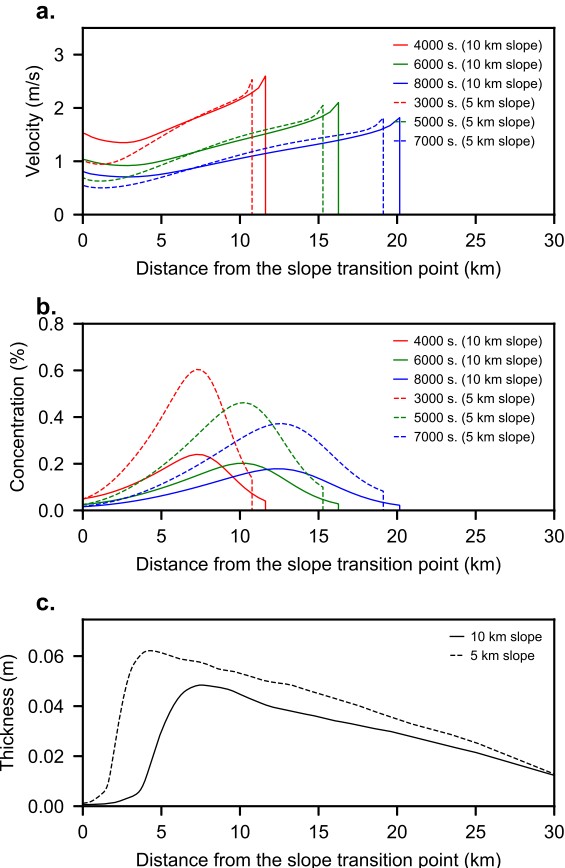

**Figure 12.** Influence of the length of upstream slope to the result of inverse analysis. 10 km long slope was used to produce a turbidite, and the bed was analyzed by the inverse model trained with 5 km long upstream slope. Solid lines are values for currents produced the bed, and the dashed lines are reconstructed values. a. Time development of flow velocity. b. Total sediment concentration. c. Distribution of bed thickness.

the turbidites. The inverse analysis of hydraulic conditions is possible because the depositional characteristics are sensitive to conditions of turbidity currents. The self-acceleration of turbidity flow is an extreme example of the sensitivity of turbidites to the flow initial conditions.

## 5.2  Applicability to field-scale problems

To apply this method to outcrops, the extent of the area that should be surveyed to collect data and the interval between outcrops should be determined. The tests with different sizes of sampling windows suggest that the survey region should be located more than 10 km from the proximal region (Fig. 5). The loss function (i.e., the MSE of the estimates of the parameters) decreases as the length of the sampling window increases, and the best result is obtained at the 10 km-long window. Regarding the interval

of the outcrops, the test results of sampling rates of more than 1.0% with interpolation for data at non-sampled grids are not inferior to the full sample. Since the training data used in this study are computed on 5 m-spaced grids, extracting data from these grids with a 1.0% probability is equivalent to conducting an inverse analysis from outcrop data that are distributed at 0.5 km intervals on average. Although the RMSEs of the model prediction certainly increase when the sampling rate decreases below 1.0 %, the RMSE values does not drastically worsen until 0.5 %. Therefore, even if the outcrop spacing is about 1 km, it should be possible to obtain a reasonable estimates of the flow characteristics.

These requirements for accurate inversion are attainable in the actual field. For example, Hirayama and Nakajima (1977) correlated individual turbidites of the Pleistocene Otadai Formation distributed in the Boso Peninsula, Japan, on the basis of the key tuff beds. Their correlation covered a region over 30 km long with 33 outcrops. Thus, the average interval between outcrops was approximately 1 km. Amy and Talling (2006) correlated individual beds in the Miocene Marnoso Arenacea Formation, Italy, using the Contessa MegaBed and an overlying "columbine" marker bed as the key beds. Their correlation covers 109 sections of approximately 30 m thick succession and extends over 120 km in a direction parallel to flow. Other studies in various regions (e.g., the Arnott Sandstone in France) also reported the correlation of individual turbidites in similar scale and frequency (Hesse, 1974; Tokuhashi, 1979, 1989; Amy et al., 2000, 2004). Furthermore, Bartolini et al. (1972) surveyed the Western Alboran Basin Plain, Mediterranean Sea, and discovered an individual turbidite on the sea floor at 49 cores over approximately 30 km. The records of cores in similar scale and intervals have also been reported by other studies of the modern submarine fans in different areas (Bornhold and Lilkey, 1971; Pilkey et al., 1980). In summary, although the method proposed in this study requires fairly high resolution data of turbidite individual beds correlated over a long distance, such conditions in ancient geological records as well as modern seafloor surveys can be achieved.

Besides these outcrop conditions, measurement errors in the field are another important factor for application. The test results suggest that the proposed inverse model of this study is very robust against random noise; random errors in the measured data have little effect on the results (Fig. 10). Therefore, even if localized and small-scale scouring and sedimentation occur due to some processes such as bottom currents after the deposition of a turbidite, results of inverse analysis will not be seriously affected. However, if deposits of multiple events are amalgamated to form a single thick massive sandstone, the hydraulic conditions reconstructed from the bed should be considerably different from the actual conditions. To avoid this situation, it is important to identify the erosional surface inside the bed carefully at the actual outcrop. In addition, it is safer not to analyze massive sandstones that are more than several meters thick, because they are likely to be amalgamated deposits.

Perhaps the most significant drawback to analyze actual turbidites is the assumption about the topography of the upstream submarine canyon. In this study, we tested doubling the length of the upstream slope and found that the predicted values for the concentration that were different from the original values (Fig. 12). In case of actual analysis, the upstream topography can be set correctly if the modern submarine fan is analyzed. Regarding ancient turbidites, however, some assumptions about the length and scale of the submarine canyons are necessary without measurements. In this case, it is recommended to set up various lengths of submarine canyons within a reasonable range, and to examine the degree to which these assumptions affect the inverse analysis results carefully. Nevertheless, it is worth noting that the test results were reasonable for velocity (Fig. 12), even if the assumption about the length of upstream slope was substantially different. This suggests that the inverse

model proposed in this study can generally reconstruct the behavior of turbidity currents in sedimentary basins, even if the
development process of turbidity currents upstream is different.

## 5.3 Comparison with previous methodologies

In existing inverse analysis methods of turbidity currents, the difference in depositional characteristics between the outputs of
the forward model and the field observation is quantified as the objective function, and the initial and the boundary conditions
of the forward model are determined by conducting optimization calculations to minimize the objective function (Nakao
and Naruse, 2017, e.g.,). This is because models of turbidity currents are generally nonlinear and are difficult to linearize,
especially when considering the entrainment of the basal sediment (Parker et al., 1986). Although the actual computational
load depends on the choice of algorithm, this type of optimization calculation generally consists of multiple steps, and each
step depends on the results of the previous calculation. Thus, the entire optimization procedure is difficult to parallelize.
For instance, the kriging-based surrogate management method (Lesshafft et al., 2011) or the genetic algorithm (Nakao and
Naruse, 2017) have been used to optimize the objective function for inversion of turbidity currents. In these methods, multiple
calculations are conducted in each calculation step (generation), and the distribution of the objective function in the parametric
space is iteratively estimated. Although the computations within each generation can be parallelized in this kind of algorithms,
the next generation's computation depends on the results of the previous generation's computation, and therefore, the entire
computation process cannot be parallelized. Thus, if the computational load of the forward model is high, the inverse analysis
takes an unrealistic amount of time.

Parkinson et al. (2017) applied the adjoint method with the gradient-based optimization algorithm. Although the differentiation of the layer-averaged model by the adjoint method greatly reduces the load of the gradient calculation, this approach
still requires an iterative calculation for optimization. Thus, the sediment entrainment process is omitted from their model.
Their model does not consider resuspension (entrainment) process of sediment, whereas suspended sand in turbidity currents
is maintained by balancing the effects of particle settling and diffusion from the bottom (i.e. entrainment). Their model only
considers advection and settling of particles, so that the suspended sediment quickly settles and be lost over short distances at
realistic flow thicknesses and concentrations. The only way to transport large amounts of suspended sediment for long distance
and to deposit thick turbidites without resuspension is to make the flow extremely thick or to suppose unusually high velocity
or concentration. This is the reason for that the extremely thick flow depth (more than 3000 m) was obtained in their results.
Their inversion method requires iterations that cannot be parallelized, so that the forward model needs to be simplified for
this purpose. In addition, gradient-based optimization tends to have problems with initial value dependency and escaping from
local optimal solutions. For this reason, the results of their inverse analysis of turbidites were quite unrealistic. In contrast,
we were able to adopt "full model" that incorporate the entrainment process of suspended sand into our model without any
problems. As a result, our inversion did not produce any anomalous reconstructions even though most of our test data exhibit
thickness and grain size distributions similar to realistic turbidites. This strongly suggests the robustness of our inverse model
and its applicability to real turbidites.

Another potential approach to optimization is the Markov Chain Monte Carlo (MCMC) method, but even with this method, repetition of the forward model calculation is unavoidable, since MCMC usually requires repetition of calculations of objective function, which cannot be parallelized, more than the order of $10^4$ time. The layer-averaged model of unsteady turbidity currents is probably not suitable for the forward models due to their computational load.

The approach proposed in this study is obviously superior to existing methods in terms of applicability to the field, as it allows computationally demanding models to be applied as forward models. The general relationship between the bed and the input parameters is learned by NN rather than adjusting the input parameters of the numerical model to reproduce the characteristics of specific individual beds. The objective function used in the training of this NN is not the difference between the features of the sediment, but the precision of the inverse analysis results themselves. The most computationally demanding part of the inverse analysis method proposed here is the generation of the training data for the NN. However, since the computations of the forward models are completely independent of each other, the generation of the training data can be conducted in parallel. Thus, our method enables us to easily prepare a large number of training data by using PC clusters, even for very computationally demanding forward models. In addition, the number of calculations required for training is not as high as other methods, specifically only approximately 3,000. It is also advantageous that the proposed method enables us to perform various tests for robustness or precision of inversion before application to field examples, because the NN outputs results of inverse analysis extremely fast. For these reasons, we consider that this study successfully generated an inverse model using the layer-averaged model for unsteady turbidity currents that can be applied to the field.

## 5.4 Limitations and future tasks

The inverse model proposed in this study has several limitations. Inevitably, the accuracy of the inverse analysis is governed by the validity of the forward model that generates the training data. The present implementation of the inverse model uses the one dimensional layer-averaged model as the forward model, but this model is likely to be applicable only to sedimentary basins that are laterally constrained or to the inside of the submarine channels. The layer-averaged model of Parker et al. (1986) used in this study has been widely accepted, but various doubts have been recently raised such as the formulation of entrainment rates of basal sediment (Dorrell et al., 2018) and ambient seawater (Luchi et al., 2018). The assumption of a lock exchange condition for the occurrence of turbidity currents may not be appropriate in some situations.

Although Luchi et al. (2018) suggested that the a single layer model may not be sufficient for considering behavior of turbidity currents maintained over long distances, it is expected that such turbidity currents do not leave turbidites and create a bypassing zone. Otherwise, the concentration in the lower layers of turbidity currents decrease, and therefore the currents stop within a relatively short distance. Thus, a two-layer model of turbidity currents is not always necessary for inversion of bed-scale turbidites. However, modeling of continuos sustained turbidity currents is necessary for inverse analysis of the development of submarine fans and channel-levee systems in a larger scale.

It is relatively easy to solve these problems described above. Without changing the framework of the proposed method, we can adapt to any situation by changing the forward model to generate the training data. For processes such as sediment transport, it is easy to revise the model to incorporate the state-of-the-art knowledge. By adopting computationally demanding models,

inverse analysis using 2-D and 3-D forward models may be possible. In Future research, these issues should be addressed, and the methodology to actual field examples should be applied.

The analysis of ancient turbidites is an important issue in the future. However, even if ancient turbidites are analyzed, it is not possible to verify that the results obtained are correct, because the hydraulic conditions for ancient turbidity currents are unknown. Another way to verify the validity of the method is to reconstruct the hydraulic conditions of experimental turbidity currents from the turbidites deposited in the flume, and compare them with the measured values. The turbidity currents measured in the modern submarine canyons and their deposits would be another candidate to be used for the model verification.

## 6    Conclusions

This study implemented an inverse model that reconstructs the flow characteristics of turbidity currents from their deposits using a NN, and verified its effectiveness at the field scale. In this study, we assumed that turbidity currents occur from suspended sediment clouds, which flow down from the steep slope in a submarine canyon to a gently sloping basin plain. The inverse model attempts to reconstruct seven model input parameters (height and length of the initial suspended sediment cloud, sediment concentration of four grain size classes, and slope of the basin plain) from the thickness and grain size distribution of the turbidite deposited on the basin plain. The forward model using one-dimensional layer-averaged equations was used to produce training data sets with random conditions in prescribed ranges. The NN was trained using the generated data to develop the inverse model. Thereafter, the test data generated independently from the training data were analyzed to verify the performance of the inverse model.

As a result of the training and tests conducted on the inverse model, the following was found:

1. More than 2000 data sets were required for the training to avoid overlearning. An increase in the number of training data sets results in improved performance of the inverse model; however, the degree of improvement becomes smaller even if more than 3000 data sets.

2. The hydraulic conditions and basin slopes were precisely reconstructed from the test data sets. The thickness and grain size distribution of the turbidites deposited over a 10 km-long interval in a sedimentary basin were sufficient to reconstruct the flow conditions.

3. The inverse model of this study is quite robust to random errors in the input data. The addition of a normal random number with about the same magnitude of the standard deviation to the original data had little effect on the results of the inverse analysis.

4. Judging from the results of subsampling tests, the inversion of turbidity currents can be performed if an individual turbidite can be correlated over 10 km at approximately 1 km intervals.

These results imply that the inverse model of turbidity currents proposed in this study is promising for analyzing field-scale turbidites. This method is expected to be applied to actual turbidites in the future.

## 7 notation

The symbols L, M and T denote dimensions of length, mass and time respectively. The symbol [1] denotes that the value is dimensionless.

$C_T$   Total layer-averaged sediment concentration [1]

     $C_i$   Layer-averaged sediment concentration of the $i$th grain-size class [1]

     $C_f$   Bed friction coefficient [1]

     $e_{si}$   Sediment entrainment coefficient [1]

     $F_i$   Volumetric fraction of the $i$th grain-size class in the active layer [1]

$L_a$   Thickness of the active layer [L]

     $R$   Submerged specific density of sediment particles $(= 1 - \rho_s/\rho_f)$ [1]

     $S$   Bed slope [1]

     $U$   Layer-averaged velocity of turbidity currents $[\mathrm{LT}^{-1}]$

     $g$   Acceleration of gravity $[\mathrm{LT}^{-2}]$

$h$   Flow depth of turbidity current [L]

     $l_0$   Initial length of suspended sediment cloud [L]

     $r_{0i}$   Ratio of near-bed sediment concentration of the $i$th grain-size class to layer-averaged concentration [1]

     $t$   Time [T]

     $w_{si}$   Settling velocity of sediment of the $i$th grain-size class $[\mathrm{LT}^{-1}]$

$x$   Bed-attached streamwise coordinate [L]

     $H_0$   Initial height of suspended sediment cloud [L]

     $J$   Number of model input parameters [1]

     $K$   Number of test datasets [1]

     $N$   Number of grain size classes [1]

$S_l$   Basin slope [1]

$\eta_T$  Thickness of the turbidite [L]

$\eta_i$  Volume per unit area of sediment of the $i$th grain-size class [L]

$\lambda_p$  Porosity of the turbidite [1]

$\rho_s$  Density of sediment particles [$\mathrm{ML}^{-3}$]

$\rho_f$  Density of the water [$\mathrm{ML}^{-3}$]

*Code and data availability.*  All codes and data used in this study is deposited in the respository Zenodo (https://doi.org/10.5281/zenodo.4135168)

*Author contributions.*  HN: Conceptualization, Methodology, Software, Writing, Reviewing and Editing. KN: Software.

*Competing interests.*  All authors declare that: (i) no support, financial or otherwise, has been received from any organization that may have an interest in the submitted work; and (ii) there are no other relationships or activities that could appear to have influenced the submitted
work.

*Acknowledgements.*  This work was supported by JSPS KAKENHI Grant Numbers 26287127 and 20H01985. This study was also supported by the Earthquake Research Institute The University of Tokyo Joint Usage/Research Program 2018-B01.

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
