# Peer review of "Inverse modeling of turbidity currents using an artificial neural network approach: verification for field application"

_Earth Surface Dynamics, 2020_

## Referee Comment (RC1) · Anonymous Referee #1 · 2 Jan 2021

Understanding the differences between relatively frequent turbidity currents observed in recent studies and the records of ancient turbidites is the motivating problem of this study. As the authors clearly state in the introduction, this is a critical, long-standing problem that must be solved to fully understand submarine sedimentation processes. In this study the authors propose the use of an artificial neural network to solve the inverse problem, i.e. to determine the characteristics of the turbidity current from the thickness and grain size distribution of the deposit. The neural network is trained using a dataset of numerically-generated turbidity currents and deposits, and the proposed inverse model returns the inlet (upstream boundary) conditions of the model runs, as well as the basin slope, i.e. the slope of the mildest reach in the model domain (Figure

3 of the submitted manuscript). This is an interesting approach to solve a very complex problem. I do not have enough knowledge to comment on the appropriateness of neural networks. However, I can comment on the forward model used to generate the numerical database and on the usefulness of the inverse model output. As I further discuss below, it is critical that the authors fully acknowledge up front (i.e. in section 1 and/or 2) the limitations of the forward model used in their analysis, and further discuss how the upstream boundary conditions may provide information on the local characteristics of the turbidity current that emplaced a certain type of deposit tens of kilometers downslope.

1) Forward model selection The authors should clearly acknowledge in the introduction or at the beginning of section 2 (Forward model description) that the four equation model (Parker. et al., 1986, Kostic and Parker, 2006) is inadequate to model the long runout turbidity currents considered in this study. In section 5.4 (Limitations and future tasks) the authors say that 'various doubts have been recently raised' on the applicability of the four equation model and this is, at best, an understatement. Luchi and co-authors (S. Balachandar, G. Seminara and G. Parker, the author of the 1986 and 2006 models) in 2018 clearly state: 'Parker et al. [1986] proposed a three−equation model and a four−equation model that formulate turbidity current dynamics in terms of layer−averaged equations. Both models treat the current as a single layer below infinite ambient fluid, with parameters depending only on the streamwise direction, x, and time, t. These equations include: a Reynolds−averaged momentum equation, a conservation equation for water, a conservation equation for suspended sediment and a layer−integrated conservation equation for energy of the turbulence. Their solution captures the streamwise evolution of layer-averaged velocity and suspended sediment concentration. However, neither model is able to explain the formation of continuous, long runout turbidity currents.' (Luchi et al., 2018). A thorough discussion on the limitations of the three and four equation models and why they should not be used is presented in the Supporting Information of the Luchi et al. (2018). The authors can say that they propose a methodology and to simplify the computational costs they use

a simplified but physically inadequate model to generate the numerical database.

2) Inverse model output Output of the inverse model are the input parameters of the numerical simulations, which are not representative of local flow conditions that result in the emplacement of a deposit with certain characteristics, e.g. thickness, sediment size distribution and internal fabric. I thus wonder why it is important to predict the upstream boundary conditions of the numerical model to 1) characterize the turbidity current and 2) understand the differences between relatively frequent turbidity currents observed in recent studies and the records of ancient turbidites. Shouldn't an inverse model tell us how the turbidity current velocity, thickness and grain size change in space and time during the event? What am I missing here? Further, the length of the steep reach in the forward model simulations does not seem to be varying from one simulation to the other (Figure 3 of the manuscript). I wonder if this has (it should) any implication on the characteristics of the turbidity current in the basin, and how this can impact the output of the forward model.

References Kostic, S. & Parker, G. (2006). The response of turbidity currents to a conyon-fan transition: internal hydraulic jumps and depositional signatures. Journal of Hydraulic Research 44 (5), 631-653. Luchi, R., Balachandar, S., Seminara, G., & Parker, G. (2018). Turbidity currents with equilibrium basal driving layers: A mechanism for long runout. Geophysical Research Letters, 45, 1518–1526. https://doi.org/10.1002/2017GL075608. Parker, G., Fukushima, Y, & Pantin, H. M. (1986). Self-accelerating turbidity currents. Journal of Fluid Mechanics 171, 145-181.

---

## Referee Comment (RC2) · Anonymous Referee #2 · 22 Feb 2021

General comments:

In this paper, the authors proposed a method for reconstructing paleo-flow condition of turbidity currents from submarine deposits using neural network technique. To obtain the data for supervising the neural network and showing the performance of the model, the authors used a layer-averaged model of turbidity currents as a forward model and performed a lot of numerical calculations regarding turbidity currents and their deposits under the different initial conditions. The supervised neural network provides a robust relationship between initial conditions of the turbidity current and resultant sediment deposit in the modeled basin. By using this method, the authors suggested that the

paleo-flow condition can be reconstructed by calculating a forward model of turbidity current with the initial conditions estimated from the deposit using the neural network. The authors also presented some implications of this method to field-scale sedimentary deposit, suggesting that the proposed method can be used to field-scale cases.

We can not get sufficiently large dataset regarding the turbidity current characteristics and their deposit in the field for supervising the neural network. Instead, physically-based numerical model might be able to provide such dataset. The presented method seems robust enough for field application. In addition, this modeling may provide something interesting things for numerical modelers. We also use forward model to reproduce the phenomena we observed and predict what will happen future. In this process, generally we perform a lot of calculation to get best result and sometimes only the best result is presented without failure cases of numerical calculations. The inverse modeling the authors did also need a lot of computational effort, but might be useful to show clear relationship of the model input parameters and output results. Overall, the paper is well fitted the scope of the Esurf and the results are interesting and clearly presented. I have some comments about the concept of the modeling and application for field cases, which need to be addressed for acceptance.

In this modeling, the authors focused on one event of turbidity current and subsequent sediment deposit and use these numerical results to supervise the neural network. But it will be possible that the sediment deposit we will sample in the field has been generated by several turbidity currents. In addition, the sediment deposits that were created by one turbidity current will be affected by many other physical processes such as erosional event by self-accelerating turbidity current and soil compaction etc, so that some information might be missing in the current condition. Is the proposed model still robust for such cases for inversion analysis? The factors I mentioned above are just small compared to the random noise the authors obtained in the analysis? Some discussion about this kind of uncertainty might be useful to show the model performance as well.

The authors mentioned about self-accelerating turbidity current in section 5.1, saying that some errors of the initial conditions caused by the inversion analysis might not be significant for estimating paleo-flow condition. I am not sure that this is also valid if the authors include some calculations of self-accelerating case. Here, the initial bed is treated as an immobile bed, so that there is no possibility of happening the self-accelerating turbidity current in this model. Since the turbidity current the authors are thinking in this paper is extremely big one, so that small differences in initial conditions might be able to cause self-accelerating turbidity current, resulting in big differences in model prediction.

As a last statement of this paper, the authors concluded that the method proposed can be applicable field-scale problem, and the application for real turbidites will be future work. Indeed, the discussion about the effect of measurement errors on the result (section 5.2) suggested that the model is robust against to such error. Also, the modeling framework of this model (section 5.3) will have some advantages to predict paleo-flow condition, which can not be reasonably reconstructed by previous studies (e.g., Parkinson et al., 2017). However, it is still not obvious that the proposed model can provide reasonable paleo-flow condition from real-turbidites. I guess that the performance of this model is highly dependent on the forward model. Since Parkinson et al. (2017) used similar types of layer-averaged model and failed to give reasonable result, I suspect that proposed model also gives such unrealistic result even though some optimization method has been improved in the present model. Some further discussion might be useful for understanding the model performance.

Lastly, I am not sure about the most important contribution of this study. The author's team already performed similar analysis, i.e., Mitra et al. (2020) for tsunami case, so that it is not clear that this study proposed new method and apply it to the turbidity current case, or the authors just applied the model, which is already proposed, to the turbidity current case. A brief introduction of the significance of this study will be merit.

Line-by-line comments:

Line 7: I feel that term "shallow water equation" should be used for river flow model, instead, for turbidity current case, "Layer-averaged model" will be better expression (actually, throughout the paper, the authors use both). This should be consistent, although these models are mathematically very similar.

Line 11: I am not sure the number "3500" has specific meaning. Is it small or large number?

Lines 279-280: What is the reason of unstable behavior of the result when the sampling window is shorter than 5 km? In addition, does it mean that detailed field measurement with the less than 1km spatial sampling window does not improve or help for inversion?
* * *

---

## Author Comment (AC1) · 28 Feb 2021

**Reply to Reviewer1**

Thank you very much for your thoughtful comments. Our replies to your comments are as follows. We will revise the manuscript to incorporate all of these discussions.

[Figure]

Comment 1: necessity of two layer model

Thank you for your important comment. As mentioned in section 5.4, in a turbid current flowing over hundreds of kilometers, the upper layer of the current is predicted to be continuously diluted while the lower layer remains highly concentrated, thus maintaining the current over long distances. Existing one-layer shallow water equation models are insufficient to reproduce such phenomena. The forward model of this study is not an exception.

However, the focus of this study is on rapidly decelerating sedimentary turbidity currents. Normally graded turbidites are considered to be deposited from such decaying flows. In this study, the distribution of turbidites is assumed to be limited to about several tens km at most, and the separation of the lower and upper layers that occurs in sustained turbidity currents after flowing tens of kilometers does not need to be considered when calculating such relatively small-scale turbidity currents. In fact, Kostic and Parker (2006), on which the forward model of this study is based, has been verified to reproduce turbidity currents at experimental and small natural scales (e.g., Fildani et al., 2006). This suggests that the inverse model in this study is well suited to analyze a single bed of turbidites that generally exhibit normal grading in strata.

It is expected that turbidity currents maintained over long distances do not leave turbidites and create a bypassing zone. Otherwise, the concentration in the lower layers of turbidity currents decrease, and therefore the currents stop within a relatively short distance. Thus, a two-layer model of turbidity currents is not always necessary for inversion of bed-scale turbidites. However, modeling of continuos sustained turbidity currents is necessary for inverse analysis of the development of submarine fans and channel-levee systems in a larger scale. This is an interesting research topic in the future. We will fully describe this aspect in the Introduction and Section 2 of our revised manuscript.
Comment 2: the reason why the inverse model reconstructs initial conditions

As you pointed out, our objective of the inverse analysis is the local conditions of a turbidity current (velocity, concentration, etc.) that resulted in a turbidite. That is why we used inverse analysis to estimate the initial conditions of the flow. This is because, as shown in Figure 9, we can obtain the state of the turbidity current at any location and time by calculating the time evolution of the forward model from the initial conditions. On the other hand, if we conduct inverse analysis to reconstruct all the values of the local flow conditions at a certain location, we have to estimate an almost infinite number of parameters because the flow velocity and concentration at the point of interest keep changing with time. If we consider all the hydraulic conditions at any given time as independent parameters, the number of values to be obtained in the inverse analysis will be the same as the number of time steps in the forward model. To avoid this situation, we decided to reconstruct the initial conditions at the onset of the flow. In this way, we can obtain the behavior of the flow with a relatively small number of parameters. This approach has already been tried successfully by Lesshafft et al. (2011), and Falcini et al. (2009) also reconstructed flow conditions of turbidity currents by obtaining boundary conditions of the model. Thus, the approach we adopted here is a standard procedure in this research field.

This is an essential point for the inverse model in this study, but we acknowledge that it has not been fully explained yet in the current text. In the revised manuscript, we will explain these points in the sections of Introduction, Forward model, Inverse model, and Discussion. We will also add a new figure in the revised paper to illustrate our strategy as described above. In that figure, the reconstructed results of the temporal changes of velocity and concentration at several locations with the original data to exhibit that our method can estimate the flow conditions at a particular location.

Comment 3: length of slope

In this study, the length of the slope located upstream was not changed. The slope from the point of flow occurrence to the bottom works as a hypothetical generator of turbidity currents. The hydraulic conditions on the slope do not necessarily have to be realistic because the role of the slope is to allow the sediment cloud to acquire the structure of a turbidity current and the flow conditions on the slope are not subjects to be estimated. If the length of the slope is changed, the flow initial conditions on the slope may vary. However, our aim is to obtain sufficiently realistic values of the velocity, concentration and thickness of the flow on the sedimentary basin. In other words, if the hydraulic conditions at the upstream end of the basin are the same, we do not need to worry about the differences in velocity and thickness of the flow on the slope caused by differences in slope length. To clarify that the slope length is not significant in this study, we will conduct new numerical experiments to show the effect of the slope length on the results of the inverse analysis in the revised manuscript.

---

## Author Comment (AC2) · 28 Feb 2021

**Reply to Reviewer 2**

Thank you very much for your thoughtful comments. Our replies to your comments are as follows. We will revise the manuscript to incorporate all of these discussions.

[Figure]

Comment 1: Bed amalgamation and disturbance

Thank you for pointing out a very interesting issue. Our inverse model produced robust results when the artificial data was subjected to quite large noise. Therefore, even if localized and small-scale scouring and sedimentation occur due to some processes such as bottom currents after the deposition of a turbidite, results of inverse analysis will not be seriously affected. However, if deposits of multiple events are amalgamated to form a single thick massive sandstone, the hydraulic conditions reconstructed from the bed will be considerably different from the actual conditions. To avoid this situation, it is important to identify the erosional surface inside the bed carefully at the actual outcrop. In addition, it is safer not to analyze massive sandstones that are more than several meters thick, because they are likely to be amalgamated deposits. These precautions will be described in the Discussion section of the revised manuscript.

Comment 2: Self-acceleration

Thank you for your interesting comment. Actually, it is in the case of self-acceleration that the inverse analysis of the initial conditions becomes easier. The relationship between turbidity currents and characteristics of turbidites is nonlinear. Especially when the flow is self-accelerating, a small difference in the initial conditions can result in very different sedimentary characteristics. This means that it is easy to find the initial conditions of the flow by inverse analysis, because even if the characteristics of the deposits are very different, the initial conditions of the flow should converge to narrow range. Thus, the inverse results in this case are expected to be robust even if there are some measurement errors in characteristics of deposits. In other words, there is a tradeoff between the robustness of the forward and inverse modeling.

This property of the inversion can be understood when we consider the opposite case. If the initial conditions of the flow are different but the characteristics of the turbidites are

exactly the same, it is impossible to estimate the flow conditions from the turbidites. The inverse analysis of hydraulic conditions is possible because the depositional characteristics are sensitive to conditions of turbidity currents. The self-acceleration of turbidity flow is an extreme example of the sensitivity of turbidites to the flow initial conditions. I will explain about this issue in section 5.1 of the revised paper.

Comment 3: Applicability to field scale problem

The applicability of the method to actual turbidites is a main topic of this paper. First, it is unlikely that our model will have the same results as Parkinson et al. (2017) because their model has an essential difference from ours, and this is the unequivocal reason why their inversion results were not realistic. Their model does not consider resuspension (entrainment) process of sediment, while suspended sand in turbidity currents is maintained by balancing the effects of particle settling and diffusion from the bottom (i.e. entrainment). Their model only considers advection and settling of particles, so that the suspended sediment quickly settles and be lost over short distances at realistic flow thicknesses and concentrations. The only way to transport large amounts of suspended sediment for long distance and to deposit thick turbidites without resuspension is to make the flow extremely thick or to suppose unusually high velocity or concentration. This is the reason for that the extremely thick flow depth (more than 3000 m) was obtained in their results. Their inversion method requires iterations that cannot be parallelized, so that the forward model needs to be simplified for this purpose. Our inverse model, on the other hand, can completely parallelize the forward model calculations, and we do not need to obtain an analytical solution for the gradient of the objective function unlike their method. Therefore, we were able to adopt "full model" that incorporate the entrainment process of suspended sand into our model without any problems. As a result, our inversion did not produce any anomalous reconstructions even though most of our test data exhibit thickness and grain size distributions similar to realistic turbidites. This strongly suggests the robustness of our inverse model and

its applicability to real turbidites.

Of course, the analysis of ancient turbidites is an important issue in the future. However, even if ancient turbidites are analyzed, it is not possible to verify that the results obtained are correct, because the hydraulic conditions for ancient turbidity currents are unknown. Another way to verify the validity of the method is to reconstruct the hydraulic conditions of experimental turbidity currents from the turbidites deposited in the flume, and compare them with the measured values. The turbidity currents measured in the modern submarine canyons and their deposits would be another candidate to be used for the model verification. However, before proceeding to these steps, we suggest that it is important to thoroughly examine the validity of the method using artificial data.

We have already described some of these points described above in the discussion section, and will describe them in more detail in the revised manuscript.

Comment 4: Novelty in methodology

This paper has a methodological novelty in that it achieves inversion of unsteady and nonuniform flows. Our research group was the first to develop a neural network based inversion method for event deposits, and the first application of this framework was the inversion of tsunami deposits by Rimali et al. (2020). This is described in the introduction section. However, the forward model used in their study was based on the assumption of quasi-steady flow, and thus our work is the first time to perform the inverse analysis using a neural network with completely unsteady flow. In addition, there are various differences in the properties of tsunamis and turbidity currents. In the case of turbidity currents, the amount of suspended sediment is linked to the driving force of the flow, while in the case of tsunamis, the two are independent of each other. Therefore, an increase in the concentration of suspended sand does not affect the flow dynamics of tsunamis. Because of these differences, there was no guarantee that the inverse analysis of turbidites would give good results, even if a similar inverse
analysis framework was used for tsunami deposits. The success of the inverse analysis for turbidity currents, which exhibit quite different properties from those of tsunamis, indicates the wide applicability of our inversion framework for event deposits.

Line-by-line comments

Line 7

We agree to unify the terminology to "layer averaged model".

Line 11

We used the number 3500 to mean a relatively small number. However, for the sake of clarity, we will remove this sentence in the revised paper. Instead, we will revise the sentence in line 8 as follows:

*A reasonable number (3,500) of repetition of numerical simulation using one-dimensional shallow water equations under various input parameters generates a dataset of the characteristic features of turbidites.*

Line 279–280

*We do not fully understand why the results are not stable for sampling windows shorter than 5 km, but it probably indicates that the training results fall into a local optimum solution depending on the initial values of the weight coefficients of the neural network (given by random numbers) due to incomplete information. In any case, the loss function is very good (less than 0.01), so that even turbidites that can be tracked for less than 5 km are likely to give good results if the outcrop spacing is sufficiently narrow and*
*detailed observation of beds is possible.*

*Interactive comment on Earth Surf. Dynam. Discuss., https://doi.org/10.5194/esurf-2020-93, 2020.*

---

## Author Response (AR2)

**Reply to Editor**

I thank you very much for accepting the publication of our manuscript in Earth Surface Dynamics. I changed the title of the manuscript to "Inverse modeling of turbidity currents using an artificial neural network approach: verification for field application". Also, I incorporated the comment from the reviewer to change the x axis labels of Figures 4 and 9 as "Distance from the slope transition point". I have uploaded the files with all the fixes. If there are any other corrections, I will respond as soon as possible.